# Novel Potential Markers of Myofibroblast Differentiation Revealed by Single-Cell RNA Sequencing Analysis of Mesenchymal Stromal Cells in Profibrotic and Adipogenic Conditions

**DOI:** 10.3390/biomedicines11030840

**Published:** 2023-03-10

**Authors:** Olga Grigorieva, Nataliya Basalova, Maksim Vigovskiy, Mikhail Arbatskiy, Uliana Dyachkova, Maria Kulebyakina, Konstantin Kulebyakin, Pyotr Tyurin-Kuzmin, Natalia Kalinina, Anastasia Efimenko

**Affiliations:** 1Institute for Regenerative Medicine, Medical Research and Education Center, Lomonosov Moscow State University, Lomonosovsky Ave., 27/10, 119192 Moscow, Russia; basalova.natalia@gmail.com (N.B.); vigovskiy_m.a@mail.ru (M.V.); konstantin-kuleb@mail.ru (K.K.); efimenkoan@gmail.com (A.E.); 2Faculty of Medicine, Lomonosov Moscow State University, Lomonosovsky Ave., 27/1, 119991 Moscow, Russia; algenubi81@mail.ru (M.A.); dyachkovauliana@gmail.com (U.D.); coolebyakina@gmail.com (M.K.); tyurinkuzmin.p@gmail.com (P.T.-K.); n_i_kalinina@mail.ru (N.K.)

**Keywords:** single-cell transcriptome, multipotent mesenchymal stromal cells, myofibroblast, gene expression, fibrosis

## Abstract

Mesenchymal stromal cells (MSCs) are the key regulators of tissue homeostasis and repair after damage. Accumulating evidence indicates the dual contribution of MSCs into the development of fibrosis induced by chronic injury: these cells can suppress the fibrotic process due to paracrine activity, but their promoting role in fibrosis by differentiating into myofibroblasts has also been demonstrated. Many model systems reproducing fibrosis have shown the ability of peroxisome proliferator-activated receptor (PPAR) agonists to reverse myofibroblast differentiation. Thus, the differentiation of multipotent cells into myofibroblasts and adipocytes can be considered as processes that require the activation of opposite patterns of gene expression. To test this hypothesis, we analyzed single cell RNA-Seq transcriptome of human adipose tissue MSCs after stimulation of the myofibroblast or adipogenic differentiation and revealed several genes that changed their expression in a reciprocal manner upon these conditions. We validated the expression of selected genes by RT-PCR, and evaluated the upregulation of several relevant proteins using immunocytochemistry, refining the results obtained by RNA-Seq analysis. We have shown, for the first time, the expression of neurotrimin (NTM), previously studied mainly in the nervous tissue, in human adipose tissue MSCs, and demonstrated its increased gene expression and clustering of membrane receptors upon the stimulation of myofibroblast differentiation. We also showed an increased level of CHD3 (Chromodomain-Helicase-DNA-binding protein 3) in MSCs under profibrotic conditions, while retinol dehydrogenase-10 (RDH10) was detected only in MSCs after adipogenic induction, which contradicted the data of transcriptomic analysis and again highlights the need to validate the data obtained by omics methods. Our findings suggest the further analysis of the potential contribution of neurotrimin and CHD3 in the regulation of myofibroblast differentiation and the development of fibrosis.

## 1. Introduction

Myofibroblasts are considered the main effector cells in the development of fibrosis. They secrete an excessive amount of extracellular matrix (ECM) proteins such as collagens, fibronectin, proteoglycans, and contract ECM by attaching through focal adhesions and intracellular stress fibers. One of the key markers of myofibroblasts is alpha smooth muscle actin (αSMA), which is incorporated into stress fibers [1,2]. The specific composition and rigidity of tissue ECM formed by myofibroblasts can regulate the differentiation of surrounding cells and lead to the progression of fibrosis [3,4,5,6]. The source of myofibroblasts is considered as mainly tissue fibroblasts and other stromal cells, however, the pool of myofibroblasts can also be replenished by circulating fibrocytes originating from the bone marrow, the transdifferentiation of epithelial and endothelial cells, and even macrophages [7,8]. In addition, recent studies have indicated mesenchymal stromal cells (MSCs) as one of the sources of myofibroblasts [9].

MSCs were first discovered by Friedenstein in 1974 in bone marrow. These cells turned out to be a component of the hematopoietic stem cell (HSC) niche, being an absolutely necessary participant in hematopoiesis [10]. In addition, MSCs were found to be multipotent—with the ability to differentiate into osteocytes, chondrocytes, and adipocytes. Later, cells with the same characteristics were found in many organs and tissues and isolated in culture. The perspectives of using MSCs as a source of cellular material for therapy were largely due to the multipotency of these cells, however, convincing evidence of the incorporation of exogenously introduced MSCs into the body and their differentiation into functional elements of the tissue could not be found. The current scientific paradigm explains the positive effect of MSCs on the course of tissue regeneration by the specific paracrine activity of these cells, which determines their proangiogenic, neuroprotective, antiapoptotic, and immunomodulatory effects [11,12,13]. In addition, a number of studies have shown the inhibitory effect of MSCs on the development of fibrosis [14,15,16]. Thus, MSCs can be considered as one of the key regulators of homeostasis in different tissues in normal conditions and during the development of a pathological process.

The minimum criteria for defining MSCs were formulated back in 2006. They include adhesiveness to plastic, a certain immunophenotype (CD73+/CD90+/CD105+/CD45-/CD34-/CD14-/CD11b-/CD79α-/CD19-/HLA-DR-), and the potential for differentiation in three directions [17]. However, at present, the demand to revise the definitions of MSCs has become urgent. In recent years, data have been accumulated that convincingly indicate the differences between MSCs isolated from different tissues as well as the great heterogeneity of MSCs even within the same tissue. The development of lineage-tracing technologies has made it possible to trace the origin and fate of a rather heterogeneous cell population, defined by researchers as “MSCs”. In particular, it has been shown that some cells with the formal criteria of MSCs can differentiate into myofibroblasts. The data obtained require further research, since they create a certain contradiction in the interpretation of the role of MSCs in the development of fibrosis. In our study, we analyzed the single-cell transcriptome of human MSCs cultured under profibrotic conditions to detect changes in the gene expression in a cell population capable of differentiating into myofibroblasts. In addition, we identified genes that changed the level of expression in MSCs in reciprocal directions under profibrotic or adipogenic conditions. Of note, it is known that inducers of peroxisome proliferation activating receptors (PPARs) are capable of stimulating myofibroblast dedifferentiation. Our data made it possible to identify a number of genes whose expression was increased upon the induction of myofibroblast differentiation but decreased upon the induction of adipogenesis. The obtained results suggest novel possible markers of myofibroblasts and expand the understanding of the mechanisms of MSC involvement in fibrosis.

## 2. Materials and Methods

### 2.1. Human MSC and HDF Isolation and Culture

Human adipose-derived MSCs and human dermal fibroblasts (HDF) were isolated from subcutaneous adipose tissue obtained from four healthy donors during abdominal surgery using enzymatic digestion [18]. All donors gave their informed consent and the local ethics committee of the Medical Research and Education Center of Lomonosov Moscow State University (IRB00010587, Moscow, Russia) approved the study protocol (#4, 04 June 2018). These cells were CD45-/CD73+/CD105+/CD90+/NG2+/PDGFRβ+ [11]. Primary MSCs were cultured in AdvanceSTEM™ Mesenchymal Stem Cell Media containing 10% AdvanceSTEM™ Supplement (HyClone, Logan, UT, USA) and 1% antibiotic–antimycotic solution (HyClone) at 37 °C, 5% CO_2_. Primary HDF were cultured in DMEM low glucose (Gibco) containing 10% fetal bovine serum (FBS, Gibco) and 1% antibiotic–antimycotic solution (HyClone) at 37 °C, 5% CO_2_. Cells were passaged at 70–80% confluency using Versene solution (Paneco) and HyQTase solution (HyClone).

### 2.2. Single-Cell Droplet-Based RNA-Seq Library Preparation and Sequencing

For profibrotic conditions, MSCs of 2–4 passages were seeded on decellularized extracellular matrix (dECM) obtained from human dermal fibroblasts (HDF). HDF of 4–8 passages were seeded at 20,000 cells per cm^2^ density and cultured for 14 days with the addition of 0.1 mg/mL ascorbic acid (Sigma, St. Louis, MO, USA) to stimulate ECM deposition. dECM samples were prepared as described in Novoseletskaya et al. (2020) [19]. MSCs seeded on dECM were treated with 5 ng/mL TGFβ-1 (Cell Signaling) and cultured for 96 h (F). For adipogenesis induction, MSCs on plastic were treated with growth medium containing 10 µM dexamethasone, 10 µM insulin, 200 µM indomethacin, and 0.5 mM 3-isobutyl-1-methylxantine (Invitrogen, Waltham, MA, USA) for 96 h (Ad). Control cells were cultured in standard conditions for 96 h (Control) (Figure 1). The single cell suspensions of MSCs were converted to barcoded scRNA-Seq libraries using the Chromium Next GEM Single Cell 3′ GEM (10× Genomics), aiming for 10,000 cells per library. Samples were processed using Library & Gel Bead Kit v3.1 barcoding chemistry (10× Genomics). Single samples were processed in a single well of a PCR plate, allowing all cells from a sample to be treated with the same master mix and in the same reaction vessel. Samples were processed in parallel in the same thermal cycler and Illumina HiSeq1500 sequencing system (Illumina, Inc., San Diego, CA, USA).

### 2.3. Analysis and Quality Control of Single-Cell RNA-Seq Data

Samples were mapped to the reference genome (human reference genome NCBI build 38, GRCh38) using CellRanger 6.1.2 (10× Genomics). We used the following quality control criteria: cells with <2500 or >7500 detected genes or <7000 or >70,000 RNA counts or over 5% of unique molecular identifiers (UMIs) derived from the mitochondrial genome were filtered out as low-quality cells. Data from samples were processed using R-studio 1.4 with R 4.1.2 and Seurat 4.0.4 regressing out mitochondrial genes [20]. The integration of datasets were performed using Seurat function IntegrateData. Principal component analysis of the integrated datasets was performed on the variable genes, and 20 principal components were used for cell clustering (resolution = 0.3) and UMAP dimensional reduction. The cluster markers were found using the FindAllMarkers function. Cell types were manually annotated based on the cluster markers using g:GOSt functional profiling.

### 2.4. cDNA Libraries Quality Analysis

The cDNA concentration in the samples was analyzed by Qubit using the Qubit DYNA HS Assay Kit (Thermo Fisher Scientific, Waltham, MA, USA, Q32851). The quality of the obtained libraries was evaluated on a Bioanalyzer 2100 using a set of High Sensitivity DNA Kit reagents (Agilent Technologies, Santa Clara, CA, USA, 5067-4626). The quality of all obtained libraries complied with the requirements for the samples sent for sequencing.

### 2.5. Library Preparation, Sequencing, and Alignment

We used from 4 to 10,000 living cells for further analysis. Using these cell samples according to the manufacturer’s protocol of a commercial kit for 10× Genomics, we prepared libraries for high-performance sequencing. Sequencing of the paired-end library prepared on the Chromium (10× Genomics) device was carried out on HiSeq1500 (Illumina) using the Chromium Next GEM Single Cell 3′ GEM, Library & Gel Bead Kit (10× Genomics), with a reading length of 150 nucleotides. The average number of readings per sample was 300 million. The depth of readings was 1.2 billion readings pairs. The raw data were mapped to the human genome (version hg 38) using CellRanger count (v. 6.0.0).

### 2.6. RNA-Seq Data Bioinformatic Analysis

The fastq-files were processed using a software CellRanger count to receive the .bam, .cloupe, and .aggr files. Cells with fewer than 10,000 UMI readings and containing mitochondrial genes were excluded from further analysis using CellRanger reanalyze mode. For automatic cell typing, a Single R-package (celldex package) was used (according to cellular references, a Human Primary Cell Atlas (HPCA) and Blueprint). To determine the protein localization in the cell, we used data from the UniProt database. The CellMarker and PanglaoDB databases were used for manual typing. For the clustering of highly represented genes by GO:BP, we used the g:Profiler, STRING, and online platform ShinyGO v0.741: ShinyGO v0.741: Gene Ontology Enrichment Analysis + more for human genes with a P-value cutoff oof 0.05. The Cell Ranger–Loupe Browser was used for visualization.

### 2.7. Isolation of Total RNA

The total RNA isolation from the cell lysates was carried out using a commercial RNeasy Mini Kit (Qiagen, Germany) according to the manufacturer’s protocol. The total RNA concentration was determined using a NanoDrop 1000 spectrophotometer (Thermo Scientific, Waltham, MA, USA) with the original ND-1000 V 3.7.1 software (Thermo Scientific, Waltham, MA, USA). Samples with absorption ratios at wavelengths of 260 and 280 nm (A260/280) from 1.9 to 2.1 were used for further analysis.

### 2.8. Real-Time PCR with Reverse Transcription

cDNA synthesis was performed using a commercial MMLV RT Kit (Eurogen, Russia), according to the user manual. Amplification was performed using the Nexus Mastercycler^®^ gradient device (Eppendorf, Germany). Quantitative real-time PCR was performed using the qPCRmix-HS SYBR + LowROX Kit (Eurogen, Russia) in accordance with the manufacturer’s protocols on a QuantStudio™ 5 Real-Time PCR System (Applied Biosystems, Waltham, MA, USA). Real-time PCR data were analyzed using the ΔCT method to evaluate the expression of the main genes normalized for the housekeeping gene (36b4) in dynamics. The 2^−ΔΔCT^ method was used to evaluate the expression level of the target gene in the experimental samples compared to the untreated control samples. The sequences of the primers used are shown in Table 1.

### 2.9. Immunocytochemistry

MSCs were fixed with 4% paraformaldehyde solution (Panreac) at room temperature for 10 min and incubated with 0.2% Triton ×100 (Sigma, St. Louis, MO, USA) solution at RT for 10 min (except neurotrimin labelling). Furthermore, MSCs were incubated for 1 h in 1% bovine serum albumin (BSA, Sigma) and 10% normal goat serum (Abcam, Cambridge, UK) solution at room temperature to block the non-specific interaction of antibodies. Subsequently, the samples were incubated with primary polyclonal rabbit antibody for αSMA (Biolegend, San Diego, CA, USA, 904601), perilipin (Thermo Fisher Scientific, PA1-1051), CHD3 (Cloud-Clone Corp., Wuhan, China, PAA317Mu01), neurotrimin (Affinity Biosciences, Melbourne, Victoria, Australia, DF4245), RDH10 (Affinity Biosciences, DF12105), or rabbit polyclonal IgG (Biolegend, 910801) in 1% BSA solution at +4° overnight. Then, the samples were incubated with fluorescence-labeled goat anti-rabbit or goat anti-mouse (Invitrogen, A-11001) secondary antibodies (A11034, Invitrogen) at room temperature for 1 h. Cell nuclei were labeled with DAPI (DAKO, Glostrup, Denmark). Samples were analyzed with a Leica DM6000B fluorescent microscope equipped with a Leica DFC 360FX camera (Leica Microsystems GmbH, Wetzlar, Germany) using the LasX program. The percentage of CHD3+ MSCs was evaluated in FIJI using IgG-based thresholding.

### 2.10. Statistical Analysis

Statistical data processing was carried out using GraphPad Prism 9 (Version 9.4.1). An unpaired *t*-test was used to verify the reliability of differences in the data between the experimental and control groups. The differences were considered statistically significant at *p* < 0.05.

## 3. Results

### 3.1. Cultured MSCs Respond Differently to Profibrotic Stimuli

When cultivating MSCs on dECM derived from dermal fibroblasts with the addition of TGFβ-1 for 96 h (F), the number of cells including αSMA in the stress fibers increased in culture, which represents a classic attribute of myofibroblast differentiation (Figure 2A). According to the CellMarker and PanglaoDB databases, myofibroblasts are characterized by the expression of a number of genes: TNS1—tensin-1; CDH11—cadherin-11; PALLD—palladin, cytoskeletal associated protein, CALD1—caldesmon-1, TAGLN—transgelin, MYL9—myosin light chain 9, ACTA2—alpha-smooth muscle actin, DES—desmin, and GFAP—glial fibrillary acidic protein. However, when analyzing the transcriptome of single cells of the entire population, we failed to isolate the population of MSCs that responded to TGFβ-1 stimulation by an increase in the expression of all of the listed genes. We identified myofibroblasts by the increased expression of ACTA2 (log24), MYL9 (log27), and TAGLIN (log27), thus defining a cluster of differentiated cells (Figure 2C). Thus, MSCs respond unequally to stimulation with profibrotic stimuli.

At the same time, upon the stimulation of the adipogenic differentiation of the total MSC population, we observed that some cells acquired the expression of perilipin-2 (PLIN2) after 96 h (Figure 2B), while the population retaining αSMA+ stress fibers significantly decreased. Perilipins, proteins associated with intracellular lipid droplets, are conventional markers of adipogenic differentiation. Thus, we hypothesized that the induction of adipogenesis leads to the activation of the expression of genes suppressed during myofibroblast differentiation. For further comparison, we used MSCs that differentiate into myofibroblasts under profibrotic conditions (F) or adipocytes under adipogenic conditions (Ad) (Figure 2D,E).

### 3.2. Identification of Genes That Change Expression in MSCs in Opposite Directions upon Stimulation of Myofibroblast and Adipogenic Differentiation

A total of 2159 genes were upregulated in two F samples: 702 of them increased by more than 1.5 times; 2985 genes were downregulated, 997 of them decreased by more than 1.5 times. In the two Ad samples, 1955 upregulated genes were found, of which 1095 changed by more than 1.5 times and 2534 downregulated genes, of which 1739 changed by more than 1.5 times.

In two biological repeats, the number of all upregulated in F but downregulated in Ad genes was 729, 63 of them changed by more than 1.5 times in both samples, and the number of downregulated in F but upregulated in Ad genes was 755, 55 of which changed by more than 1.5 times in both samples. The list of the 63 and 55 genes was shortened to eight and three genes, respectively, selected because they had the same pattern in both repeats.

Figure 2 shows the genes with differential expression changes in two repetitions of the total analyzed populations (D) and when comparing the isolated population of myofibroblasts with induced adipogenesis (E).

For two used biological replicates, a detailed analysis of gene expression changing in opposite directions revealed eight genes whose expression increased in MSCs under profibrotic conditions while decreasing in MSCs under adipogenic conditions (COL8A1, FN1, PRICKLE1, NTM, UCHL1, CHD3, VCAN, MEG3). Three genes increased the expression in MSCs under adipogenic conditions while decreasing in MSCs under profibrotic conditions (PLIN2, PDLIM1, MT1X). For all genes in the first group, in the subsequent analysis by RT-PCR, we showed a significant decrease in expression during adipogenic induction compared to the control. For the NTM, RDH10, VCAN, PRICKLE1, and UCHL1 genes, significantly different expression was shown during adipogenic induction compared to the induction of myofibroblast differentiation.

### 3.3. Gene Expression Analysis in MSCs by RT-PCR

We validated the detected changes in gene expression by real-time PCR with reverse transcription. The expression of the genes in MSCs stimulated with profibrotic stimuli was compared with MSCs after the induction of adipogenesis after 4 days of exposure; MSCs under conditions of serum deprivation served as controls. The obtained values were normalized to the control to exclude the effect of variability in response between cells from different donors.

The analysis confirmed an increase in the expression level of extracellular matrix protein genes upon the stimulation of myofibroblast differentiation such as FN1, COL8A1, and VCAN. Upon the stimulation of adipogenesis, the expression of the mentioned genes significantly decreased compared to the control. However, there was no significant multidirectional change in expression for the most of the genes selected by the analysis of scRNA-Seq using RT-PCR (Figure 3) in the total MSC population.

### 3.4. Analysis of Protein Markers in MSCs

We analyzed the synthesis level of two proteins, CHD3 and NTM, which changed the expression in both the total population of MSCs under profibrogenic conditions and in an isolated myofibroblast cluster under adipogenic conditions. Even though we were not able to see significant differences in the level of mRNA, we were able to show an increase in the number of CHD3+ cells when cultured under profibrogenic conditions compared to the adipogenic ones by immunocytochemistry (*p* = 0.07604, Figure 4). We discovered that CHD3 was accumulated in the nuclei of MSC, which may indicate its activation.

Furthermore, we showed for the first time that the expression of neurotrimin in human adipose tissue MSCs increases after the induction of myofibroblast differentiation. According to the immunocytochemistry results, neurotrimin tended to cluster on the cell membrane in profibrogenic conditions compared to both the control group and the adipogenic differentiation group (*p* = 0.03381, Figure 4A,C and Appendix A).

Changes in the expression of the RDH10 gene were shown only when a cluster of myofibroblasts was isolated and compared with MSCs upon the stimulation of adipogenesis. Despite the fact that PCR analysis showed a significant decrease in the expression of the RDH10 gene under adipogenic conditions compared with the stimulation of myofibroblast differentiation, we were able to detect RDH10 only in MSCs after the stimulation of adipogenesis by ICC (Figure 4).

## 4. Discussion

The primary cell culture, defined as “MSC”, possesses significant heterogeneity, which is observed as differences in cell morphology, the expression of surface antigens, and the response to various stimuli [21,22]. Only a part of the MSC population can be considered multipotent stem cells capable of in vitro trilineage differentiation into osteogenic, chondrogenic, and adipogenic directions. The ability of tissue-specific MSCs and MSC-like cells to turn into myofibroblasts and contribute to organ fibrosis has also been demonstrated in multiple studies [9,23]. Our search for factors makes it possible to separate the subpopulations of MSC-like cells during differentiation induction, resulting in the discovery of genes not previously characterized as involved in myofibroblast differentiation. This allows us to suggest new and clarify the existing mechanisms of fibrosis development.

During wound healing, TGFβ-1 is a key factor stimulating myofibroblast differentiation. Binding to the TGFΒRII receptor, TGFβ-1 induces its dimerization and subsequent phosphorylation of the SMAD2/3 or SMAD1/5/9 mediators, which then form a complex with SMAD4 and translocate into the nucleus [24,25]. This pathway, known as canonical, leads to the upregulation of the characteristic genes of myofibroblasts—αSMA, EDA-FN, type I collagen, and others [26]. The fully differentiated myofibroblast has αSMA-rich stress fibers that enable it to contract surrounding tissues, contributing to wound closure during healing [1,2,26]. Furthermore, myofibroblasts produce a vast amount of ECM, which forms an intercellular environment for further tissue repair [26,27]. Normally, after the successful completion of repair, an excess amount of myofibroblasts is eliminated through apoptosis. In cases of pathology, myofibroblasts are able to accumulate, synthesizing an excessive amount of ECM with a composition different to that of normal tissue ECM [28,29]. It has been shown that the increased stiffness and profibrotic composition of ECM promote the differentiation of new myofibroblasts, potentiating the effect of TGFβ-1 [30,31].

Today, various methods to control myofibroblast differentiation are being actively investigated. One promising approach includes studying the adipogenic differentiation, since the adipogenic and myofibroblast directions are somewhat mutually exclusive ways of differentiation. For instance, it has been shown that reduced activity of PPARγ, a master regulator of adipogenic differentiation, was observed in many fibrotic diseases and correlated with the ability of the fibroblasts to acquire traits of the myofibroblast phenotype. Conversely, PPARγ agonists prevented the differentiation of fibroblasts into myofibroblasts in vitro and reduced fibrosis severity in vivo [32]. Accumulating data on the functional heterogeneity of MSCs suggest that the total population of MSCs contains populations with different abilities to differentiate into adipocytes and myofibroblasts. Thus, a comparative study of these subpopulations could allow us to reveal new ways to regulate cell differentiation to myofibroblasts.

In our study, we found that genes upregulated during the myofibroblast differentiation of MSCs encode proteins involved in cell adhesion, ECM organization, and morphogenesis. We observed that the induction of adipogenic differentiation led to the upregulation of genes involved in the cell response to glucocorticoids and calcium signaling (Appendix A).

Notably, only for the PRICKLE1 gene was a significant increase in expression upon the stimulation of myofibroblast differentiation shown. We assumed this to be due to the heterogeneity of the MSC population only a part of which differentiated into myofibroblasts. To test this hypothesis, we refined the bioinformatics analysis by excluding MSCs that did not respond to profibrogenic conditions with a significant increase in the expression of myofibroblast characteristic genes such as ACTA2, TAGLIN, and MYL. By this, we compared differentially expressed genes in a cluster of myofibroblasts isolated from the general population and in MSCs with induced adipogenesis. The obtained method allowed us to reveal other genes whose expression increased during myofibroblast differentiation: SERPINE2, MYO1B, LIMS2, SLIT3, MINDY2, RDH10, AVEN, CLIP1, GARS, and TPM2. However, validation by RT-PCR also did not reveal significant differences between the expression of these genes in the control and in MSCs under profibrogenic conditions. Perhaps such validation requires a subpopulation of the total MSC population prior to PCR analysis.

We also found that COL8A1, PRICKLE1, and MT1X became excluded from the list of differentially expressed genes after the isolation of the myofibroblast cluster. Thus, it can be concluded that these genes are upregulated in MSCs that do not respond/respond slowly to profibrotic conditions by differentiation into myofibroblasts. Collagen VIII is a non-fibrillar short-chain collagen that has been shown to be involved in vasculogenesis processes including the migration and maintenance of the phenotype of smooth muscle cells [33]. The role of collagen VIII, as a multicellular substrate facilitating the migration of endothelial cells during angiogenesis, intimal invasion of smooth muscle cells, and myofibroblasts migration during fibrotic conditions, has been discussed [34,35]. However, it should be noted that the absence of collagen VIII reduces the ability of fibroblasts to TGFβ-1 induced differentiation into myofibroblasts and the development of cardiac fibrosis in collagen VIII knock-out (col8KO) mice in an aortic banding model [36].

Prickle planar cell polarity protein 1 (PRICKLE1) is a core component of the non-canonical Wnt/planar cell polarity pathway. By participating in this signaling pathway, Prickle promotes the polarization of cells (for example, chondrocytes [37]) and makes a significant contribution to the differentiation of osteoblasts [38]. Upregulation of PRICKLE1 in MSCs that do not respond/weakly respond to profibrotic conditions may point to the cell population committing to the osteo- and chondrogenic direction in response to dECM. In addition, PRICKLE1 regulates cell migration, contributing to the disassembly of focal contacts [39,40], most likely by binding to the RICTOR protein [41]. Thus, the decrease in the PRICKLE1 expression in cells during the induction of adipogenic differentiation may be associated with the loss of the ability of cells to migrate as they differentiate in the adipogenic direction.

Metallothionein 1X(a product of the MT1X gene) is a small membrane-bound protein of the Golgi apparatus that protects the cell from toxic metal ions and oxidative stress. Expression of the MT1X gene is regulated, in particular, by the transcription factor USF-1 [42]. USF-1 activates the expression of fatty acid synthesis genes in response to insulin [43], which is observed during adipogenic differentiation. Therefore, the increase in MT1X expression in MSCs during the induction of adipogenic differentiation is presumably a consequence of the work of the transcription factor USF-1.

To analyze the revealed differently expressed factors at the protein level in MSCs after the stimulation of differentiation into myofibroblasts and adipocytes, we selected proteins that were not previously characterized in the processes of MSC differentiation—CHD3 and neurotrimin. CHD3 (Chromodomain-helicase-DNA-binding protein 3), an enzyme encoded by the CHD3 gene, is one of the components of a histone deacetylase complex called the Mi-2/NuRD complex, which is involved in chromatin remodeling. Our data indicate that the stimulation of myofibroblast differentiation results in both an increase in CHD3 gene expression in MSCs and an increase in the number of CHD3+ cells, while CHD3 is found in the cell nuclei. Since it can participate in both gene silencing and in chromatin “activation”, the mechanism of its participation in the processes of myofibroblast differentiation needs further study [44,45].

Neurotrimin, a member of the IgLON family of proteins, is a nerve cell adhesion molecule that promotes neurite outgrowth in DRG neurons through heterophilic and homophilic interactions [46]. There is evidence of neurotrimin expression in cells of mesenchymal origin. Transcriptome analysis of the total population of human bone marrow MSCs revealed NTM1 expression [47], however, the authors note that the transcriptome data require further validation. Omics studies have shown that the level of neurotrimin is increased in stellate cells of the liver [48]. NTM1 is upregulated in the lung fibroblasts of patients with idiopathic pulmonary fibrosis compared to the lung fibroblasts of healthy donors [49], while the level of the neurotrimin protein is reduced in rat bone marrow MSCs during adipogenic differentiation [50]. However, these results were not confirmed by PCR or other methods. Importantly, we have shown for the first time the expression of neurotrimin in human adipose tissue MSCs both at the mRNA and protein level, and suggest its possible contribution into MSC myofibroblast differentiation. We observed neurothrimin clustering on the MSC membrane in response to profibrogenic stimuli. However, the role of neurotrimin in myofibroblast differentiation remains to be elucidated.

Despite the fact that the data of the transcriptome and PCR analysis indicated an increase in the content of the RDH10 transcript during MSC myofibroblast differentiation, we showed through immunocytochemistry that the RDH10 protein was synthesized mainly under conditions of adipogenesis stimulation in a subpopulation of MSCs. In recent years, there has been evidence pointing to the existence of regulatory subpopulations within MSCs that can regulate the processes of differentiation of the general population of MSCs through the production of paracrine factors. We observed that, under profibrotic conditions, RDH10 gene expression increased in one of the MSC subpopulations. Since the enzyme retinol dehydrogenase, encoded by the RDH10 gene, is necessary for the formation of retinoic acid from retinol [51], a paracrine regulator of cell differentiation processes, it can be assumed that this subpopulation of MSCs could be one of the regulatory ones.

## 5. Conclusions

The data obtained in our study indicated the heterogeneity of the cultured MSCs, as only a part of the cells responded to profibrotic stimuli by differentiating into myofibroblasts. We revealed a panel of genes oppositely activated in MSCs during adipogenic and myofibroblast differentiation evaluated by scRNA-Seq. The obtained omics data allowed us to pay attention to the increased expression of neurotrimin in MSCs cultured in profibrotic conditions. Further study revealed changes in neurothrimin clustering on the surface of the cell membrane, which were not previously shown during MSC differentiation into myofibroblasts. An increased expression of neurotrimin, previously shown in samples of organ fibrosis, can be considered as one of the mechanisms that regulate the differentiation of myofibroblasts. We also observed the translocation of the CHD3 factor into the nucleus of MSCs during differentiation into myofibroblasts, which may reveal the mechanism of epigenetic regulation of the differentiation process. However, the transcriptome analysis data did not match the protein analysis data for RDH10, as this factor was increased upon adipogenic differentiation. Taken together, we revealed some novel potential markers of myofibroblasts originating from MSCs and suggest the contribution of neurotrimin and CHD3 in the regulation of myofibroblast differentiation and the development of fibrosis. These factors could serve as potential targets to control the development of fibrosis as well as to adjust the properties of MSCs as a drug for regenerative medicine. It should be noted that scRNA-Seq transcriptomic changes in differentiating MSCs only partially reproduced the expression profile when validated by RT-PCR. Thus, our data also once again highlights the need to validate the results obtained by omics methods to draw more reliable conclusions regarding the molecular changes in cells and their functional properties.

## Figures and Tables

**Figure 1 biomedicines-11-00840-f001:**
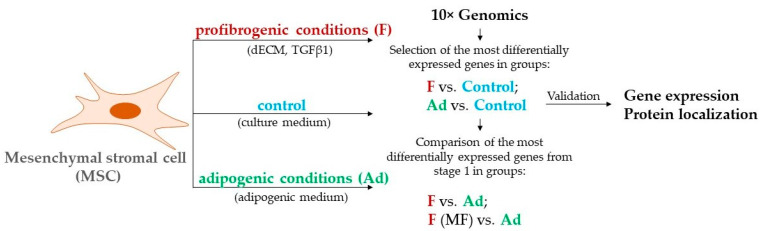
Experimental design included single-cell RNA-Seq (10× Genomics) of human adipose-derived mesenchymal stromal cells (MSC) in profibrotic conditions (F) or adipogenic conditions (Ad) compared to the control conditions. The genes that changed the expression in these samples were compared between the F and Ad samples of MF (myofibroblast cluster from F sample) and Ad.

**Figure 2 biomedicines-11-00840-f002:**
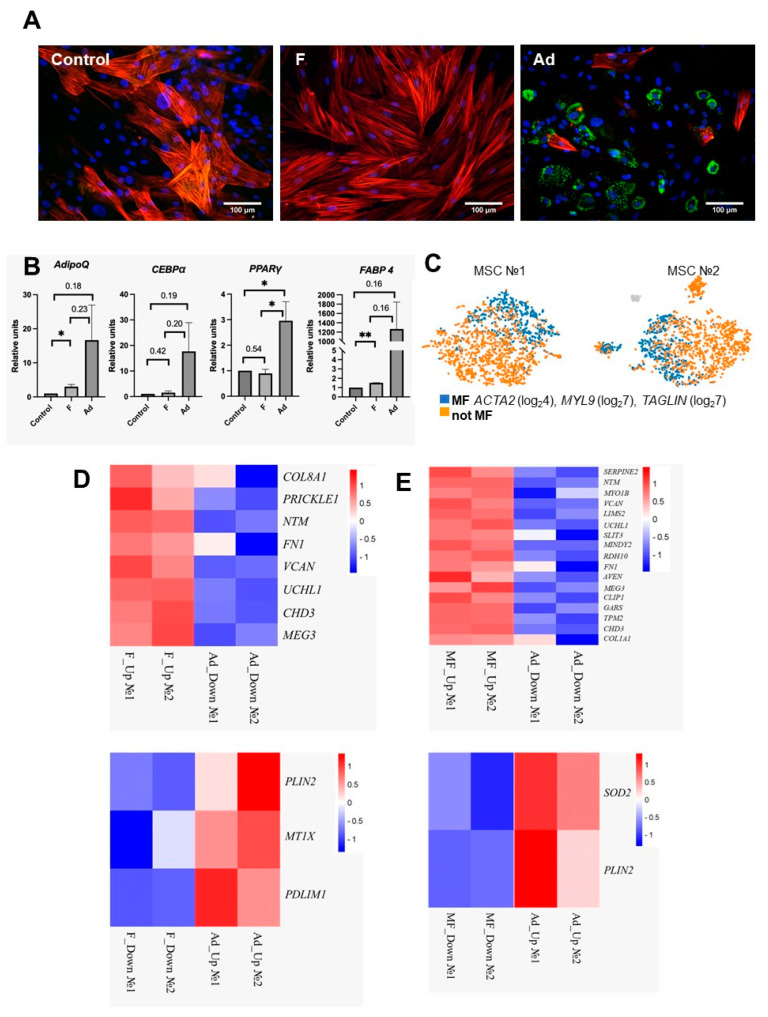
(**A**) Expression of αSMA (red) and perilipin (green) in cultured human MSCs after 4 days of TGFβ-1 stimulation (F) or the induction of adipogenesis (Ad); nuclei are labeled with DAPI. (**B**) Analysis of changes in the expression of genes characteristic of adipogenesis in MSCs, mean ± SE, * *p* < 0.05, ** *p* < 0.01, with *p* > 0.05 *p* value is indicated. (**C**) Isolation of a cluster of myofibroblasts in the total analyzed population, MF—myofibroblasts. (**D**) Genes that change expression in MSCs in different directions after 4 days under profibrogenic conditions (F) or adipogenic (Ad) (**E**) Genes that change expression in different directions in MSCs after 4 days in profibrogenic conditions in an isolated cluster of myofibroblasts (MF) or adipogenic (Ad).

**Figure 3 biomedicines-11-00840-f003:**
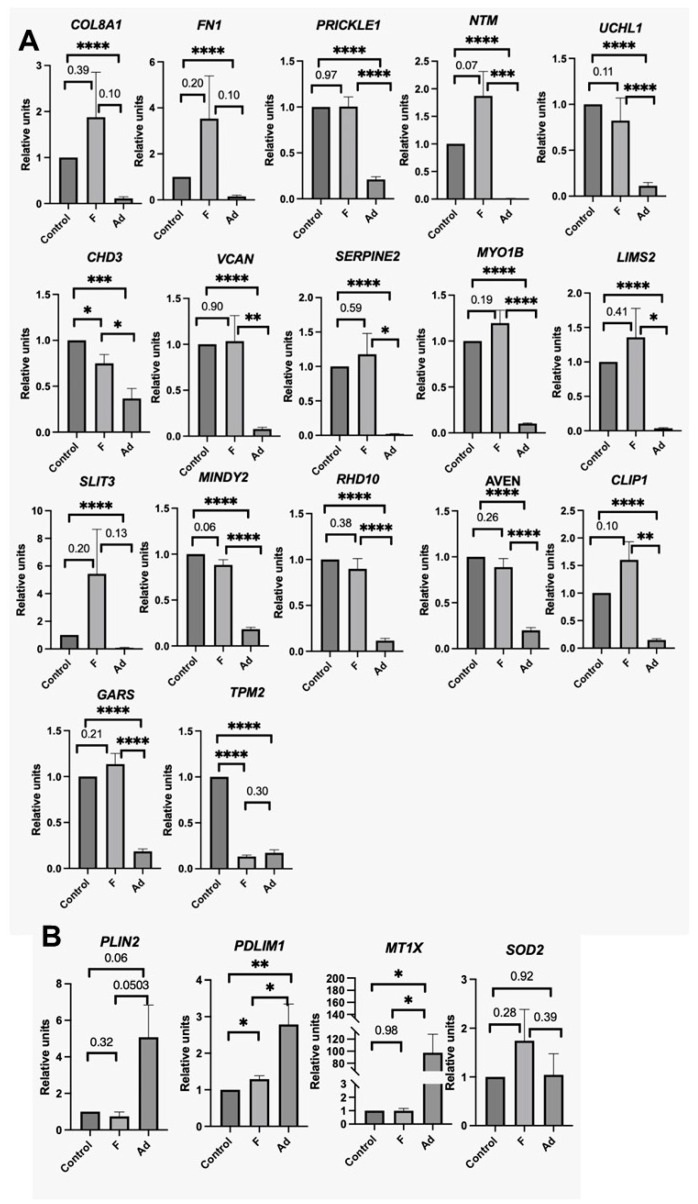
Change in gene expression evaluated by RT-PCR. The data are presented as the ratio of the level of gene expression to the control, mean ± SE, * *p* < 0.05, ** *p* < 0.01, *** *p* < 0.001, **** *p* < 0.0001, at *p* > 0.05 indicates *p* value. (**A**) Genes whose expression increased upon differentiation into myofibroblasts and decreased upon induction of adipogenesis according to transcriptome analysis, (**B**) Genes whose expression decreased upon differentiation into myofibroblasts and increased upon the induction of adipogenesis, according to transcriptome analysis.

**Figure 4 biomedicines-11-00840-f004:**
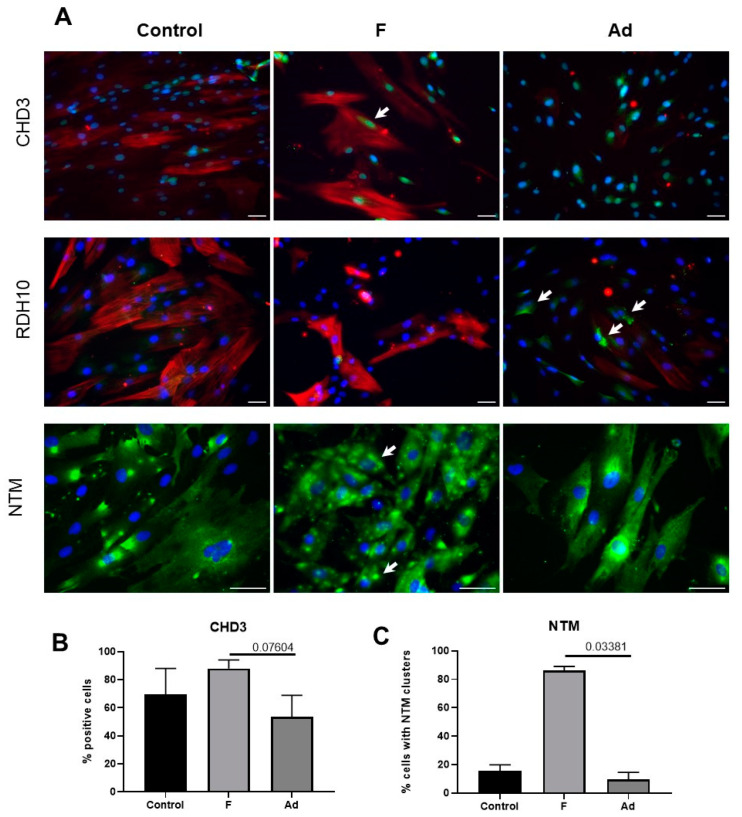
Immunocytochemical evaluation of CHD3 (green), RDH10 (green), NTM (green), and αSMA (red) in the control (Control) MSCs and after incubation in profibrotic (F) or adipogenic conditions (Ad). Fluorescent microscopy, blue staining—nuclei stained with DAPI, scale bar is 50 µm (**A**). The percentage of CHD3-positive cells, mean ± SE, *p* value is indicated (**B**). The percentage of cells with neurotrimin clusters, mean ± SE, *p* value is indicated (**C**).

**Table 1 biomedicines-11-00840-t001:** Primer pairs used for the qRT-PCR and the respective gene names.

Gene	Protein	Forward	Reverse
*36B4*	Acidic ribosomal phosphoprotein P0	5′-GCTGCTGCCCGTGCTGGTG-3′	5′-TGGTGCCCCTGGAGATTTTAGTGG-3′
*ADIPOQ*	Adiponectin	5′-GACCAGGAAACCACGACTCA-3′	5′-TTTCACCGATGTCTCCCTTAGG-3′
*PPARG*	Peroxisome proliferator-activated receptors	5′-TCAGGTTTGGGCGGATGC-3′	5′-TCAGCGGGAAGGACTTTATGTATG-3′
*CEBPA*	CCAAT—enhancer-binding protein alpha	5′-TATAGGCTGGGCTTCCCCTT-3′	5′-AGCTTTCTGGTGTGACTCGG-3′
*FABP4*	Fatty acid binding protein 4	5′-ACTGGGCCAGGAATTTGACG-3′	5′-CTCGTGGAAGTGACGCCTT-3′
*CHD3*	Chromodomain-helicase-DNA-binding protein 3	5′-CCGTCAGCATTGGGTGTGAA-3′	5′-TCTTGCGTTTTCGGGGTTTTC-3′
*NTM*	Neurotrimin	5′-CCAAAGACCTCTAGGGTCCAC-3′	5′-GTCTCCAAGTAACCGTAGGCT-3′
*RDH10*	Retinol dehydrogenase 10	5′-ACCTGACGGCTGAAAGAGTC-3′	5′-GAAAAGCCTTAGTGGTCCAGAAG-3′
*FN1*	Fibronectin-1	5′-CGGTGGCTGTCAGTCAAAG-3′	5′-AAACCTCGGCTTCCTCCATAA-3′
*VCAN*	Versican	5′-GTAACCCATGCGCTACATAAAGT-3′	5′-GGCAAAGTAGGCATCGTTGAAA-3′
*COL8A1*	Collagen Type VIII Alpha 1 Chain	5′-GGGAGTGCTGCTTACCATTTC-3′	5′-AGCGGCTTGATCCCATAGTAG-3′
*PLN2*	Perilipin-2	5′-ATGGCATCCGTTGCAGTTGAT-3′	5′-GGACATGAGGTCATACGTGGAG-3′
*SERPINE2*	Serpine 2	5′-TGGTGATGAGATACGGCGTAA-3′	5′-GTTAGCCACTGTCACAATGTCTT-3′
*TPN2*	Tropomyosin 2	5′-AGACCCGAGCAGAGTTTGC-3′	5′-TGGTGAATCTCGACGTTCTCC-3′
*AVEN*	Caspase and apoptosis activator inhibitor	5′-GCGCCGGTTGAAGATGACA-3′	5′-TGCAGAGCTAAGGAGGACACT-3′
*CLIP1*	CAP-Gly domain-containing binding protein 1	5′-AGGAAGGTGCAAGCAGAAGAT-3′	5′-GTTTTTGTAAGGTTGCTGATCGG-3′
*GARS*	Glycyl-tRNA synthetase 1	5′-ATGGAGGTGTTAGTGGTCTGT-3′	5′-CTGTTCCTCTTGGATAAAGTGCT-3′
*LIMS2*	LIM zinc finger domain containing protein 2	5′-GCACCGGCACTATGAGAAGAA-3′	5′-ACGGGCTTCATGTCGAACTC-3′
*MINDY2*	MINDY lysine 48 deubiquitinase 2	5′-TTGCACAAACTACAGACAGGC-3′	5′-TGAGGGTCTACTAACCACCCA-3′
*MT1X*	Metallothionein 1×	5′-AACTCCTGCTTCTCCTTGCC-3′	5′-GCTCTATTTACATCTGAGAGCACAA-3′
*MYO1B*	Myosin 1b	5′-CGGATGAAGCATACAGATCCC-3′	5′-CTGCCACATAGGACATGACAAG-3′
*PDLIM1*	PDZ and LIM domain 1	5′-GCTGGCCTCTACTCTTCTGAA-3′	5′-GCTGAGCATGGTCTAAGGGT-3′
*PRICKLE1*	Prickle planar cell polarity protein 1	5′-GCTGCCTTGAGTGTGAAACG-3′	5′-TGCCCGTCATAGGTCATCTGT-3′
*SLIT3*	Slit- guidance ligand 3	5′-GGCATCGTCGAAATACGCCTA-3′	5′-GCTGATGTCTATTCGCTTCAGTT-3′
*UCHL1*	Ubiquitin carboxy-terminal hydrolase L1	5′-AATGTCGGGTAGATGACAAGGT-3′	5′-GGCATTCGTCCATCAAGTTCATA-3′
*SOD2*	Superoxide dismutase 2	5′-TTTCAATAAGGAACGGGGACAC-3′	5′-GTGCTCCCACACATCAATCC-3′

## Data Availability

Data will be shared by the lead contact upon request.

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
