# Peer review of "Novel Potential Markers of Myofibroblast Differentiation Revealed by Single-Cell RNA Sequencing Analysis of Mesenchymal Stromal Cells in Profibrotic and Adipogenic Conditions"

_biomedicines, 2023, doi:10.3390/biomedicines11030840_

Round 1

Reviewer 1 Report

The manuscript entitled "Novel potential markers of myofibroblast differentiation revealed by single-cell RNA sequencing analysis of mesenchymal stromal cells in profibrotic and adipogenic conditions" authored by Olga Grigorieva et al. describe about novel potential markers of myofibroblasts originating from MSCs and suggest the contribution of neurotrimin and CHD3 in the regulation of myofibroblast differentiation and the development of fibrosis. Overall, the study was well-conducted, technically sound and neatly presented. Just few comments to the authors.

1. While methods and results were self-explanatory for the experts, for the sake of beginners and to draw attention to more readers, it could be better if the authors include couple of schematics on (i) overall experimental approach followed in the study, (ii) mechanistic overview of the key signaling observed in the study.

2. Include a separate conclusions section and detail the major conclusions point by point if possible.

3. Fig 3 legend: scale bar mentioned was 50 "mkm". Does that mean μm?

Author Response

The manuscript entitled "Novel potential markers of myofibroblast differentiation revealed by single-cell RNA sequencing analysis of mesenchymal stromal cells in profibrotic and adipogenic conditions" authored by Olga Grigorieva et al. describe about novel potential markers of myofibroblasts originating from MSCs and suggest the contribution of neurotrimin and CHD3 in the regulation of myofibroblast differentiation and the development of fibrosis. Overall, the study was well-conducted, technically sound and neatly presented. Just few comments to the authors.

We are grateful for all the revisions and reasonable comments and suggestions. We considered all the comments.

  1. While methods and results were self-explanatory for the experts, for the sake of beginners and to draw attention to more readers, it could be better if the authors include couple of schematics on (i) overall experimental approach followed in the study, (ii) mechanistic overview of the key signaling observed in the study.

We appreciate the suggestion for improving our data presentation to make it understandable for wider audience. Therefore, we added graphical abstract (see page 2) with the main findings of the paper as well as Figure 1 with the graphical presentation of the research design. We extended the discussion section with overview of the key signaling observed in the study (see page 13).

  1. Include a separate conclusions section and detail the major conclusions point by point if possible.

Thank you for the suggestion, we agree that conclusion section will help the readers to understand the observed data better. We added section “Conclusion” to the paper (see page 15).

  1. Fig 3 legend: scale bar mentioned was 50 "mkm". Does that mean μm?

We thank the reviewer for this note, we corrected figure 4 (previously figure 3) caption «mkm» for «μm» and also figure 1S caption the same way.

Reviewer 2 Report

In present study, authors investigated probability of MSCs participating in producing fibrosis.

Loading fibrosis matrix resulting of activation of myofibroblasts in inflammatory diseases is a source of loading fibrotic tissue which make difficult regular function of organs, ex:  a fibrotic liver.

In this study researchers investigated potentiality of differentiation of MSCs to myofibroblast and has a role in production of fibrosis.

Comments:

A huge number of tests have been performed, in DNA, RNA and protein expression level in this article, however it remains a major question does not reply, how these new data can ameliorate medical treatment for fibrosis or other pathology which related to myofibroblasts. Why it was necessary to find new marker for this subpopulation.

Author Response

In present study, authors investigated probability of MSCs participating in producing fibrosis.

Loading fibrosis matrix resulting of activation of myofibroblasts in inflammatory diseases is a source of loading fibrotic tissue which make difficult regular function of organs, ex:  a fibrotic liver.

In this study researchers investigated potentiality of differentiation of MSCs to myofibroblast and has a role in production of fibrosis.

Comments:

A huge number of tests have been performed, in DNA, RNA and protein expression level in this article, however it remains a major question does not reply, how these new data can ameliorate medical treatment for fibrosis or other pathology which related to myofibroblasts. Why it was necessary to find new marker for this subpopulation.

We thank the reviewer for the careful reading of our manuscript and a very pertinent comment. To make the answer to the major question more clearly, we added the section “Conclusion” to the manuscript (see page 15). However, we would like to explain some points that turned out to be questionable. Our data suggest the contribution of such proteins as neurotrimin and Chromodomain-helicase-DNA-binding protein 3 (CHD3) in myofibroblast differentiation. These potential novel markers could be used to regulate the differentiation of new myofibroblasts, which are the main cell type leading to the progression of fibrosis. Our findings provide novel insights into the possible management of their expression in mesenchymal cells to develop new medical strategies to prevent the progression of fibrotic process in damaged tissues – a relevant approach for many severe disease treatments. In addition, the study of MSC differentiation could provide a theoretical basis for developing ways to control the properties of cultured MSCs, since this cell type is one of the most popular in the development of cell products for regenerative medicine.

Reviewer 3 Report

This article describes the research to find potential markers of myofibroblast originating from MSCs. The authors showed an increased level of CHD3 (Chromodomain-Helicase-DNA-binding protein 3) in MSCs under profibrotic conditions. Interestingly, they also observed the expression of neurotrimin in human adipose tissue MSCs; it was increased after induction of myofibroblast differentiation. These data suggested the potential contribution of neurotrimin and CHD3 in regulating myofibroblast differentiation and developing fibrosis. Their findings are beneficial to detect myofibroblast differentiation, but as the author mentioned, I agree they need to validate the results obtained by omics methods to make more reliable conclusions. Hopefully, as the authors repeatedly noted, their transcriptome data validation strengthens this study. I hope they had it at this point. 

I recommend publishing this article after minor revision; some points I noticed are the following.

1) I understand Biomedicine doesn't require a conclusion section (optional), but it is better to have a separate one. The others have 432-437, which could be the "Conclusion" section.

2) Overall, I found many sentences too long and difficult to follow, particularly in the Discussion session. For example, 341-346 and 411-416 have one sentence. It could be easier to read if the author could divide sentences by clarifying what they want to say. 

3) From lines 232-241, It is not easy to read because there are mixed-ups in commas with periods.  

For example, 1,5 times must be 1.5 times. Also, it is confusing in lines 238 and 239, and I recommend adding "and" between two numbers. 

"……. in Ad genes was 729, and 63 of them changed by more than 1.5 times in both samples. The number of downregulated in F but upregulated in Ad genes was 755, and 55 of them changed by more than 1.5 times in both samples."

4) The lists of the 63 and 55 genes were shortened to 8 and 3 genes, respectively, selected because they had the same pattern in both repeats.

5) A period missing, reference [41]. (Line 384)

Author Response

This article describes the research to find potential markers of myofibroblast originating from MSCs. The authors showed an increased level of CHD3 (Chromodomain-Helicase-DNA-binding protein 3) in MSCs under profibrotic conditions. Interestingly, they also observed the expression of neurotrimin in human adipose tissue MSCs; it was increased after induction of myofibroblast differentiation. These data suggested the potential contribution of neurotrimin and CHD3 in regulating myofibroblast differentiation and developing fibrosis. Their findings are beneficial to detect myofibroblast differentiation, but as the author mentioned, I agree they need to validate the results obtained by omics methods to make more reliable conclusions. Hopefully, as the authors repeatedly noted, their transcriptome data validation strengthens this study. I hope they had it at this point. 

I recommend publishing this article after minor revision; some points I noticed are the following.

1) I understand Biomedicine doesn't require a conclusion section (optional), but it is better to have a separate one. The others have 432-437, which could be the "Conclusion" section.

Answer: Thank you for this point, we agree that conclusion section will help the readers to understand the observed data better. We added section “Conclusion” to the paper (see page 15).

2) Overall, I found many sentences too long and difficult to follow, particularly in the Discussion session. For example, 341-346 and 411-416 have one sentence. It could be easier to read if the author could divide sentences by clarifying what they want to say. 

Answer: We thank the reviewer for this comment, to make the discussion more understandable we have rewritten the section and particularly divided 341-346 and 411-416 into separate sentences. We also relocated this part into “Result” section.

3) From lines 232-241, It is not easy to read because there are mixed-ups in commas with periods.  

For example, 1,5 times must be 1.5 times. Also, it is confusing in lines 238 and 239, and I recommend adding "and" between two numbers. 

"……. in Ad genes was 729, and 63 of them changed by more than 1.5 times in both samples. The number of downregulated in F but upregulated in Ad genes was 755, and 55 of them changed by more than 1.5 times in both samples."

Answer: We thank the reviewer for pointing out such a subtle issue. We have replaced all commas in decimals with dots to make the text easier to readers.

4) The lists of the 63 and 55 genes were shortened to 8 and 3 genes, respectively, selected because they had the same pattern in both repeats.

Answer: Thank you, the sentence was corrected.

5) A period missing, reference [41]. (Line 384)

Answer: Thank you, the sentence was corrected.

Round 2

Reviewer 2 Report

Thanks for the modifications, now it is really clear.